# Sleep Health, Individual Characteristics, Lifestyle Factors, and Marathon Completion Time in Marathon Runners: A Retrospective Investigation of the 2016 London Marathon

**DOI:** 10.3390/brainsci13091346

**Published:** 2023-09-20

**Authors:** Jesse D. Cook, Matt K. P. Gratton, Amy M. Bender, Penny Werthner, Doug Lawson, Charles R. Pedlar, Courtney Kipps, Celyne H. Bastien, Charles H. Samuels, Jonathan Charest

**Affiliations:** 1Department of Psychiatry, University of Wisconsin-Madison, Madison, WI 53706, USA; 2Department of Psychology, University of Wisconsin-Madison, Madison, WI 53706, USA; 3Division of Medical Informatics, Department of Internal Medicine, University of Kansas Medical Center, Kansas City, MO 66160, USA; m648g606@kumc.edu; 4Social and Behavioral Sciences, Psychology, University of Kansas, Lawrence, KS 66045, USA; 5Faculty of Kinesiology, University of Calgary, Calgary, AB T2N 1N4, Canada; sleep4sport@gmail.com (A.M.B.); werthner@ucalgary.ca (P.W.); dr.samuels@centreforsleep.com (C.H.S.); jonathan.charest@ucalgary.ca (J.C.); 6Centre for Sleep and Human Performance, Calgary, AB T2X 3V4, Canada; doug.lawson1949@gmail.com; 7Faculty of Sport, Allied Health, and Performance Science, Twickenham, St Mary’s University, London TW1 4SX, UK; charles.pedlar@stmarys.ac.uk; 8Institute of Sport, Exercise and Health, University College London, London WC1E 6JB, UK; c.kipps@ucl.ac.uk; 9École de Psychologie, Université Laval, Québec City, QC G1V 0A6, Canada; celyne.bastien@psy.ulaval.ca; 10Faculty of Medicine, University of Calgary, Calgary, AB T2N 1N4, Canada

**Keywords:** marathon, sleep health, electronic device use, sleep tracker, orthosomnia

## Abstract

Despite sleep health being critically important for athlete performance and well-being, sleep health in marathoners is understudied. This foundational study explored relations between sleep health, individual characteristics, lifestyle factors, and marathon completion time. Data were obtained from the 2016 London Marathon participants. Participants completed the Athlete Sleep Screening Questionnaire (ASSQ) along with a brief survey capturing individual characteristics and lifestyle factors. Sleep health focused on the ASSQ sleep difficulty score (SDS) and its components. Linear regression computed relations among sleep, individual, lifestyle, and marathon variables. The analytic sample (N = 943) was mostly male (64.5%) and young adults (66.5%). A total of 23.5% of the sample reported sleep difficulties (SDS ≥ 8) at a severity warranting follow-up with a trained sleep provider. Middle-aged adults generally reported significantly worse sleep health characteristics, relative to young adults, except young adults reported significantly longer sleep onset latency (SOL). Sleep tracker users reported worse sleep satisfaction. Pre-bedtime electronic device use was associated with longer SOL and longer marathon completion time, while increasing SOL was also associated with longer marathon completion. Our results suggest a deleterious influence of pre-bedtime electronic device use and sleep tracker use on sleep health in marathoners. Orthosomnia may be a relevant factor in the relationship between sleep tracking and sleep health for marathoners.

## 1. Introduction

The population’s interest and participation in marathons has notably increased over recent decades. For example, Reusser and colleagues (2021) showed that the participation in the Berlin Marathon grew from 244 participants in 1974 to an astounding 40,641 participants in 2018 [1]. Complimentarily, a recently completed report in collaboration with the International Association of Athletics Federations (2019) suggested that 1.1 million individuals participated in marathons across 2018 [2]. As such, marathoners account for a non-trivial portion of the population. Given the current popularity and ever-growing interest of marathoners across the population, research is warranted to better understand individual characteristics and lifestyle factors that contribute to marathon runner performance to direct training guidelines and recommendations.

Previous research has implicated individual characteristics, such as age and sex, as well as lifestyle factors, such as caffeine consumption and overall diet, on endurance runner performance [3,4,5]. However, despite the well-established relationship between sleep health and athletic performance [6], the influence of sleep health on endurance runner performance is understudied, with the majority of the existing literature performed in ultramarathoners (50 km to 160 km races) [7,8,9]. Although there is likely similarity between regular marathoners and ultramarathoners [10], the unique, increasing training and competition demands of ultramarathons also likely leads to interindividual and lifestyle differences between the two running populations [11]. Thus, there is a need to advance the understanding of the influence of sleep health on running performance strictly in a sample of marathoners. Furthermore, since sleep health is likely to play a critical role in not just a marathoner’s performance but also their risk for injury, recovery, mental health, and overall well-being [6], establishing a better understanding of the role that individual characteristics and lifestyle factors play in the sleep health of marathoners is useful to identify those at heightened risk for sleep health problems and, thus, other deleterious outcomes.

This investigation was designed to address these gaps in knowledge by evaluating the relationships between sleep health, individual characteristics, lifestyle factors, and marathon completion time in a sample of the 2016 London Marathon participants. We approached this investigation with the following specific aims:Describe the sleep health characteristics of marathoners preceding marathon competition.Examine the influence of individual characteristics, including age, sex, and marathon performance based on expectation, on sleep health outcomes.Assess the influence of lifestyle factors, including sleep tracker, electronic device, alcohol, and caffeine use, on sleep health outcomes.Evaluate the relationships of sleep health and lifestyle factors with marathon completion time.

A major strength of this investigation is the ability to characterize sleep health using the Athlete Sleep Screening Questionnaire, a validated tool purposively developed to address the specific issue of poor sleep health overestimations in athletic populations [12,13]. The results of this investigation advance the understanding of sleep health in marathoners preceding competition, while also shedding insight into interindividual differences in the sleep characteristics of marathoners, as well as lifestyle factors that influence the sleep health of marathoners. Furthermore, this investigation sheds insight into the role that sleep health characteristics and lifestyle factors play on marathon completion time.

## 2. Methods

### 2.1. Data Collection, Ethical Oversight, and Analytic Sample Formation

Data was collected from participants of the 2016 London Marathon. Participants were approached directly and in-person during the event registration process over a 4-day period. Participants were provided detailed information about the investigation from a study team member, with direct communication that completion of the provided survey indicated informed consent. Respondents had the opportunity to email the researchers to obtain their results and sleep recommendations. All study procedures were approved by the St Mary’s University Ethics Committee (Twickenham, London).

Although initial data collection did not depend upon whether the marathoner completed the marathon, this study team was provided a dataset that only included data from consenting participants who completed the marathon. This initial dataset included responses from 951 marathoners, representing 2.41% of the total marathoners who started the race (N = 39,523) and 2.43% of the total marathoners that completed the race (N = 39,091). Age is a focal individual characteristic in this investigation. This was provided in the dataset in a manner mirroring the participant’s marathon age grouping, with these groups predetermined by the London Marathon organizing committee [14]. Seven age categories were present across the initial dataset: 18–39, 40–44, 45–49, 50–54, 55–59, 60–64, 65–69, and 70 and over (70+). After looking at the distribution of runners across age groups, we determined it was best to collapse the 7 groups into a categorical variable with less levels. Initially, we considered using the whole dataset to formulate a categorical variable with three levels: Young (18–39), middle-aged (40–64), and older adults (65+). However, there was a scarcity of older adults in the sample (N = 8; 0.84% of the total sample). Given this, and the fact that older adults can significantly differ from middle-aged adults in terms of sleep, lifestyle, and running ability [15,16,17], we decided to omit this small percentage of marathoners from the analytic sample rather than combine them with the middle-aged group. As such, the final analytic sample included 943 marathoners who completed the 2016 London Marathon, representing 2.39% of the total marathoners who started the race and 2.41% of the total marathoners that completed the race.

### 2.2. Collected Measures and Variable Formation: Sleep Characteristics, Individual Characteristics, Lifestyle Factors, and Marathon Completion Time

#### 2.2.1. Collected Measures

All participants completed the Athlete Sleep Screening Questionnaire (ASSQ), along with a general, brief survey, which provided all sleep characteristics and lifestyle factors, as well as the majority of individual characteristics. These are further described below. Runner performance was captured by the official marathon completion time (minutes).

#### 2.2.2. Athlete Sleep Screening Questionnaire (ASSQ): Sleep Characteristics

The ASSQ was developed specifically for use in athlete populations, to address the specific issue of poor sleep health overestimation in athletes [12]. This tool has previously been validated in a sample of elite athletes across a variety of sports [13]. To our knowledge, this questionnaire is the only existing comprehensive sleep questionnaire purposively designed to capture sleep characteristics in athletic populations. The ASSQ is a 16-item questionnaire that was developed to examine 6 key characteristics of overall sleep health: sleep quality, total sleep time, circadian preference, insomnia, sleep disturbance while traveling, and sleep disordered breathing problems. For the purpose of this investigation, we narrowed the analytic focus of the ASSQ. Specifically, we focused on the sleep characteristics captured by the ASSQ that comprise the global sleep difficulty score (SDS).

The ASSQ SDS is derived from responses across items 1, 3, 4, 5, and 6. Item 1 inquires about total sleep time (TST), with available options including 5 to 6 h, 6 to 7 h, 7 to 8 h, 8 to 9 h, and more than 9 h. Importantly, this approach to capturing TST is a limitation of the ASSQ, as it leads to an opaque decision for a responder when they have a TST on the boundary. For example, it is unclear whether 5 to 6 h or 6 to 7 h is the best option for a responder who reports 6 h of TST per night. Item 3 inquires about the individual’s satisfaction/dissatisfaction with their sleep quality (“sleep satisfaction”), with available options including *very satisfied*, *somewhat satisfied*, *neither satisfied nor dissatisfied*, *somewhat dissatisfied*, and *very dissatisfied*. Item 4 captures sleep onset latency (SOL), with available options including *15 min or less*, *16–30 min*, *31–60 min*, and *longer than 60 min*. Item 5 captures sleep maintenance issues (“sleep maintenance”), with available options including *none*, *once or twice per week*, *three or four times per week*, and *five to seven days per week*. Item 6 captures frequency of medication use, specifically for sleep (“sleep medication use”), with available options including *none*, *once or twice per week*, *three or four times per week*, and *five to seven times per week*. Items 1 and 3 are scored on a 0–4 scale, while items 4, 5, and 6 are scored on a 0–3 scale. Coding is structured so that higher scores reflect greater problem severity across the item. The summation of scores across the 5 items reflects the global SDS score. Previous research has established global SDS thresholds reflecting different levels of clinical sleep problem severity: None (0–4), mild (5–7), moderate (8–10), and severe (11–17) [13]. Scores ≥ 8 are considered representative of athletes who could benefit from further assessment and, potentially, intervention from a sleep clinician or physician.

#### 2.2.3. Individual Characteristics: Sex, Age, and Runner Performance Based on Expectation

Participants self-reported whether they were male or female on the brief, limited survey. Although we cannot be fully certain that this does not capture self-reported, binary gender, the study team determined that this is most likely to represent biological sex at birth (“sex”). Sex was dichotomized (male vs. female) for analyses. As previously expressed in the analytic sample formation component of the methodology (see Section 2.1), we collapsed 6 age groups into a dichotomized variable representing young and middle-aged adults (young vs. middle-aged adults). Young adults were defined as the 18–39 years old age group, while 40–64 years old defined the middle-aged group. We also created a dichotomized variable to represent the individual’s performance on race day based on expectation, which reflected their marathon completion time relative to their “Good For Age (GFA)” qualifying threshold [14]. This dichotomized variable compared those who completed the marathon faster than their GFA (GFA-Faster) versus those who were slower than their GFA (GFA-Slower). Lastly, marathon completion time was utilized as an outcome measure, which was formatted in minutes for statistical analyses.

#### 2.2.4. Lifestyle Factors

The general survey also included questions related to some lifestyle factors, including whether or not the responder uses a sleep tracking device (“Sleep Tracker User”), as well as their weekly frequency of electronic device use in the hour preceding bedtime (“Weekly Electronic Device Use”), weekly amount of alcohol consumption (“Weekly Alcohol Use”), and daily amount of caffeine consumption (“Daily Caffeine Use”). For the purposes of analyses, sleep tracker user was a dichotomized categorical variable (no vs. yes), while the other variables were prepared continuously, with higher scores reflecting greater use. Weekly electronic device use was scored on a 1–4 scale, caffeine use was scored on a 1–5 scale, and alcohol use was scored on a 1–6 scale. After performing focal analyses, we also dichotomized alcohol use into a “high alcohol consumption (HAC)” versus “other” variable to better clarify alcohol use relations with marathon completion time. HAC was represented by the >14 drinks per week response option (3.08% of the entire sample).

### 2.3. Statistical Analyses

Descriptive statistics were computed for the analytic sample across sleep health characteristics, individual characteristics, lifestyle characteristics, and marathon completion time.

Linear regression was employed for all other statistical analyses. Initially, univariate models evaluated relations of individual characteristics with sleep characteristics. Additional analyses explored an interaction between sex and age, with these models regressing sleep health characteristics on sex, age, and an interaction term. Univariate analyses also were performed that evaluated relations of lifestyle factors with sleep characteristics. Lastly, unadjusted and adjusted analyses were performed that assessed relations of sleep characteristics and lifestyle factors with marathon completion time. Age and sex were included as covariates in the adjusted models.

For all focal, unadjusted (univariate) regression analyses, coefficient (β), standard error (SE), *p* value, partial eta squared (_p_η^2^), a measure of effect size, and the lower limit (LL) and upper limit (UL) from the 95% confidence interval (CI) are presented. In analyses that included both unadjusted and adjusted regressions, the *p* value and _p_η^2^ from the adjusted models are also included. When relevant, means and standard deviations for groups within a categorical variable are also presented.

Data preparation and analyses for this project were performed in MATLAB© (The MathWorks, Inc., Natick, MA, USA).

## 3. Results

### 3.1. Descriptive Statistics across the Analytic Sample

Descriptive statistics for the analytic sample across individual characteristics, sleep characteristics, lifestyle factors, and marathon completion time are presented in Table 1.

The analytic sample (N = 943) was mostly male (64.5%) and young adults (66.5%). The average marathon completion time was 251 ± 53.3 min (4.18 h), with the majority of runners (78.6%) completing the marathon at a slower pace than their GFA qualifying threshold.

Based on the ASSQ SDS, 23.5% of the sample eclipsed the established referral threshold (ASSQ SDS ≥ 8) [13]. The majority of marathoners reported a sleep duration between 6 and 8 h (73.2% of sample), a sleep onset latency of <15 min (53.9% of sample), no or infrequent (1–2x per week) sleep maintenance issues (81.4% of sample), quality of sleep that is somewhat or very satisfying (62.9% of sample), and no sleep-specific medication use (94.3% of sample).

In terms of lifestyle factors, sleep tracking device use was uncommon (88.4% of sample did not use a sleep tracker) as well as high alcohol consumption (96.9% of the sample consumed ≤ 14 drinks per week). A large majority of the marathoners provided responses indicating alcohol use of ≤7 drinks per week (86.5% of sample). Furthermore, the majority of marathoners reported drinking ≤2 caffeinated drinkers per day (51.8% of sample), yet there was also a high amount of missing responses (12.7% of sample) relative to other items. Lastly, the majority of the marathoners reported using an electronic device within 1 h of bedtime every day (58% of sample).

Table 1 presents the sample characteristics for the project. Available individual characteristics include self-identified, binary gender (male or female), age group during the London Marathon, and runner performance on race day based on completion time in relation to the “Good for Age (GFA)” qualifying standards (GFA-Faster or GFA-Slower). Sleep health characteristics were derived from the Athlete Sleep Screening Questionnaire and focused on the composite Sleep Difficulty Score (SDS) as well as its associated components: total sleep time (item #1), sleep satisfaction (item #3), sleep onset latency (item #4), sleep maintenance issues (item #5), and sleep medication use (item #6). SDS severity characterization is provided based on previously determined thresholds. Available lifestyle characteristics included nightly electronic use within one hour of bedtime, whether or not the marathoner was a sleep tracker user, daily caffeine use, and weekly alcohol use. Values are presented as proportions across the entire sample, except for sample size (count) and marathon completion time (mean ± standard deviation).

### 3.2. Relations of Individual Characteristics with Sleep Characteristics

The univariate relationships between individual characteristics and sleep characteristics are presented in Table 2.

Males reported significantly lower total sleep time (β = −0.20; *p* = 0.001; 95% CI = [−0.30, −0.07]), relative to females, while females reported significantly more frequent use of sleep-related medications (β = 0.06; *p* = 0.03; 95% CI = [0.01, 0.09]), relative to males.

Statistically significant differences were observed across age groups (young vs middle-aged adults) for many sleep health characteristics. Middle-aged adults reported significantly higher overall sleep problems, as represented by the SDS comparison (β = 0.38; *p* = 0.04; 95% CI = [0.02, 0.73]). This was driven by significantly increased total sleep time (β = 0.19; *p* = 0.01; 95% CI = [0.08, 0.31]), worse sleep satisfaction (β = 0.22; *p* = 0.003; 95% CI = [0.07, 0.35]), and more frequent sleep maintenance issues (β = 0.17; *p* = 0.004; 95% CI = [0.06, 0.29]). Younger adults did report significantly longer sleep onset latency (β = −0.21; *p* < 0.0001; 95% CI = [−0.32, −0.11]).

Runners who completed the marathon at a pace slower than their GFA (GFA-Slower) were relatively comparable to GFA-Faster on sleep characteristics. However, GFA-Slower reported significantly more frequent sleep maintenance issues (β = 0.14; *p* = 0.04; 95% CI = [0.01, 0.28]), relative to GFA-Faster.

Supplementary analyses did not identify any significant interaction relationships between sex and age on sleep characteristics. The significant sex differences observed for total sleep time and medication use in univariate analyses were also observed in these multivariate analyses that controlled for age. A significant difference between sexes related to frequency of sleep maintenance issues was observed (*p* = 0.03), when controlling for age. These analyses suggested that females had significantly more sleep maintenance issues, regardless of age. The significant difference between age groups on overall sleep problems (SDS total score), as well as total sleep time and sleep satisfaction, observed in univariate analyses was not observed in these multivariate analyses when controlling for sex. Yet, the significant differences between age groups related to sleep onset latency (*p* = 0.009) and sleep maintenance (*p* = 0.003) observed in univariate analyses, translated to the multivariate analyses that controlled for sex. These supplementary analyses are presented in Appendix A.

Table 2 presents the results from the univariate regressions between individual and sleep characteristics. Individual characteristics include sex (male vs. female), age (young adults vs. middle-aged adults), and runner performance relative to “Good for Age” (GFA) qualifying standards (GFA-Faster vs. GFA-Slower). Sleep characteristics derived from the Athlete Sleep Screening Questionnaire include the sleep difficulty score (SDS) and the items that comprise the SDS: total sleep time (TST), sleep satisfaction (Satisfaction), sleep onset latency (SOL), sleep maintenance issues (Maintenance), and sleep medication use (Medication). Higher scores are reflective of more severe sleep problems. Means (standard deviation) are provided for each group across all sleep characteristics. Regression coefficient (β), standard error (SE), *p* value, partial eta squared (_p_η^2^), and the lower limit (LL) and upper limit (UL) from the 95% confidence interval (CI) are provided from each regression.

### 3.3. Relations of Lifestyle Factors with Sleep Characteristics

The univariate relationships between lifestyle factors and sleep characteristics are presented in Table 3.

Sleep characteristics were largely comparable across those who reported using a sleep tracker and those who reported not using a sleep tracker. However, those using a sleep tracker reported significantly lower sleep satisfaction (β = 0.22; *p* = 0.04; 95% CI = [0.01, 0.42]), relative to those not using a sleep tracker.

Increasing daily caffeine use was significantly associated with lower total sleep time (β = 0.10; *p* = 0.002; 95% CI = [0.04, 0.15]), while more frequent electronic device use in the hour prior to bedtime was significantly associated with longer sleep onset latency (β = 0.06; *p* = 0.03; 95% CI = [0.00, 0.11]). No significant associations were observed in relation to weekly alcohol use.

Table 3 presents the results from the univariate regressions between lifestyle and sleep characteristics. Lifestyle characteristics include sleep tracker user, electronic device use within one hour of bedtime, daily caffeine use, and weekly alcohol use. Sleep tracker user was a dichotomized variable (yes vs. no), while other lifestyle characteristics were modeled as continuous variables where higher scores reflected greater frequency of use. Weekly electronic device use (within 1 h of bedtime) is coded on a 1–4 scale, daily caffeine use is coded on a 1–5 scale, and weekly alcohol use is coded on a 1–6 scale. Sleep characteristics derived from the Athlete Sleep Screening Questionnaire include the sleep difficulty score (SDS) and the items that comprise the SDS: total sleep time (TST), sleep satisfaction (Satisfaction), sleep onset latency (SOL), sleep maintenance issues (Maintenance), and sleep medication use (Medication). Higher scores are reflective of more severe sleep problems. Regression coefficient (β), standard error (SE), *p* value, partial eta squared (_p_η^2^), and the lower limit (LL) and upper limit (UL) from the 95% confidence interval (CI) are provided from each regression. Additionally, for sleep tracker users, the means (standard deviation) are provided for each sleep health characteristic.

### 3.4. Relations of Sleep Characteristics and Lifestyle Factors with Marathon Completion Time

Table 4 presents the results from the unadjusted and adjusted analyses that assessed relations of sleep characteristics and lifestyle factors with marathon completion time. Adjusted analyses controlled for marathoner age and sex.

For sleep characteristics, we observed a significant relationship between sleep onset latency and marathon completion time in the unadjusted analyses (β = 6.86; *p* = 0.002; 95% CI = [2.49, 11.2]) that also was observed in the adjusted analyses (*p* = 0.004). This relationship associated increasing duration of sleep onset latency with a slower marathon completion time. No other relationships were observed that were significant in both unadjusted and adjusted analyses. However, significant relationships in adjusted analyses were observed for total sleep time (*p* = 0.04) and overall sleep difficulty (*p* < 0.05). These relationships associated shorter total sleep time and more severe overall sleep difficulty problems with slower marathon completion time.

For lifestyle factors, we observed a significant relationship between weekly electronic device use within one hour of bedtime and marathon completion time in the unadjusted analyses (β = 5.20; *p* = 0.006; 95% CI = [1.51, 8.88]) that also was observed in the adjusted analyses (*p* = 0.03). This relationship associated more frequent electronic device use within one hour of bedtime with a slower marathon completion time. No other relationships were observed that were significant in both unadjusted and adjusted analyses. Yet, we did observe significant associations of daily caffeine use (β = −4.48; *p* = 0.02; 95% CI = [−8.17, −0.80]) and weekly alcohol use (β = −3.49; *p* = 0.009; 95% CI = [−6.10, −0.89]) with marathon completion time in the unadjusted analyses. These results associated greater daily caffeine consumption and weekly alcohol use with faster marathon completion time. In our secondary analysis that analyzed weekly alcohol use as a categorical variable (high alcohol consumption vs. others), we observed a differential, directional relationship than what was observed when analyzed continuously. Although the relationship between categorical weekly alcohol use and marathon completion time was a statistical trend (*p* = 0.07) in the unadjusted analyses, the directionality indicated that high-alcohol-consuming marathoners had a slower marathon completion time than all others (β = −18.5).

Table 4 presents the results from the univariate and adjusted regressions of lifestyle and sleep characteristics with marathon completion time. Marathon completion time was modeled in minutes. Adjusted models controlled for runner age and gender. Sleep characteristics derived from the Athlete Sleep Screening Questionnaire include the sleep difficulty score (SDS) and the items that comprise the SDS: total sleep time (TST), sleep satisfaction (Satisfaction), sleep onset latency (SOL), sleep maintenance issues (Maintenance), and sleep medication use (Medication). Higher scores are reflective of more severe sleep problems. Lifestyle characteristics include electronic device use within one hour of bedtime, sleep tracker user, daily caffeine use, and weekly alcohol use. Sleep tracker user was a dichotomized variable (yes vs. no), while other lifestyle characteristics were modeled as continuous variables where higher scores reflected a greater frequency of use. Weekly electronic device use (within 1 h of bedtime) is coded on a 1–4 scale, daily caffeine use is coded on a 1–5 scale, and weekly alcohol use is coded on a 1–6 scale. Weekly alcohol use was also analyzed categorically (high alcohol consumption (HAC) vs. other), with HAC capturing responses of >14 drinks. Regression coefficient (β), standard error (SE), *p* value, partial eta squared (_p_η^2^), and the lower limit (LL) and upper limit (UL) from the 95% confidence interval (CI) are provided from each univariate regression, along with the *p* value and _p_η^2^ from the adjusted regressions.

## 4. Discussion

This retrospective investigation of data collected from participants in the 2016 London Marathon was designed to advance the understanding of relations between sleep health, individual characteristics, lifestyle factors, and marathon completion time in marathoners preceding competition. The ability to characterize sleep health using the validated Athlete Sleep Screening Questionnaire (ASSQ) is a major strength of this investigation, given that the ASSQ was designed specifically for athletes to address the known problem of poor sleep estimation in athletes [12,13]. To our knowledge, this is the first investigation to evaluate relations between sleep health, individual characteristics, lifestyle factors, and marathon completion time.

A multitude of interesting, significant relations between sleep health, individual characteristics, lifestyle factors, and marathon completion time were observed across the investigation. We observed significant differences between young and middle-aged adult marathoners across sleep health characteristics. Additionally, our findings implicated a relation between increasing frequency in electronic device use within 1 h of bedtime with both greater difficulties falling asleep and a longer (“worse”) marathon completion time. Furthermore, our results suggest that the difficulty/duration a marathoner experiences with sleep initiation may have an important influence on marathon completion time. Additionally, our results highlight the potential for deleterious effects from sleep tracker use on a marathoner’s perceived sleep satisfaction, which may be driven by orthosomnia, inaccurate device feedback, or both. Overall, these results advance the understanding of sleep health in elite marathoners preceding competition, while also shedding insight into key elements that influence the sleep health of marathoners as well as marathon performance. Specific findings and their implications are discussed below.

### 4.1. Nearly One-Fourth of the Sample Reported Clinically Significant Overall Sleep Difficulties

Across the 943 marathoners in our analytic dataset, 23.5% of the participants provided responses on the ASSQ that resulted in a sleep difficulty score (SDS) eclipsing the established threshold for a recommendation to a sleep clinician (SDS ≥ 8; moderate or severe SDS). Given that this is a foundational application of the ASSQ in marathoners and the ASSQ has only been applied to a few athletic samples to date, we did not approach this investigation with an *a priori* hypothesis regarding the proportion of marathoners that would eclipse this clinically significant SDS threshold. However, it is noteworthy that the proportion we observed in this sample of marathoners was similar to the proportion (25.1%) observed by Bender and colleagues (2018) in the initial validation of the ASSQ within a sample of mixed-sport elite athletes [13]. Importantly, marathoners represent a unique population of athletes that are likely vulnerable to sleep-related issues due largely to training demands [18], but also due to frequent, nagging injuries and—at times—psychological factors [19,20]. These athletes generally are not professional, which means that they often have other occupational, academic, social, and familial responsibilities that put major constraints on available hours to train across the day. Yet, training for marathons requires consistent, high-volume training loads. For example, Doherty and colleagues (2020) concluded in their recently published meta-analysis that 44 km or 4.5 h of running per week is warranted when training for a marathon completion time of 4 h [21]. This does not include additional time that may be necessary during training for recovery. Finding time for the high volume of training is a major challenge for non-professional marathoners. Unfortunately, this may result in athletes shortchanging sleep due to early AM or late PM training sessions. Furthermore, late PM training may lead to heightened pre-sleep arousal [6,22], which can translate into difficulties with sleep initiation. These factors highlight the augmented risk that marathoners are likely to face for poor sleep health. Given the critical importance of sleep health on training, recovery, performance, and well-being in athletes, it is essential that future research focuses on broadly progressing the understanding of sleep health in marathoners. Advancing this knowledge can help guide the future development of training guidelines and recommendations that appropriately account for the import of sleep.

### 4.2. Middle-Aged Marathoners Generally Reported Worse Sleep Health Than Younger Adults

In this investigation we also assessed differences in the ASSQ SDS and its components across young (18–39) and middle-aged (40–64) adult marathoners. Generally, we saw a pattern of worse sleep health characteristics in middle-aged marathoners, relative to young adult marathoners. Specifically, middle-aged adults reported significantly shorter total sleep time and worse sleep satisfaction, as well as a greater frequency of sleep maintenance issues and overall sleep difficulties. These findings may reflect changes in sleep that naturally occur throughout the lifetime, as sleep duration generally shortens and more fragmentation is common, which could result in overall worse sleep satisfaction [15,16]. Yet, those trends are often more attributed to older adults (65+). Another possibility is that these differences capture differences in occupational, familial, and/or medical/psychological factors. Perhaps middle-aged marathoners have more complicated lifestyles due to increased occupational and familial responsibilities that result in a more congested schedule, less time to integrate training, and, thus, deprioritized sleep. Furthermore, the heightened responsibilities could, in turn, translate into increased psychological stress, which may contribute to more sleep fragmentation. Similarly, the middle-aged marathoners may be more susceptible to other medical problems, such as obstructive sleep apnea and chronic pain, that are generally associated with worse sleep maintenance and satisfaction. Further research is warranted to better clarify the potential contributing factors that may be underlying the observed, generally worse sleep health characteristics in middle-aged marathoners, relative to young adult marathoners.

Although sleep health characteristics were generally worse in middle-aged marathoners, young adult marathoners did report significantly longer sleep onset latency (SOL), relative to middle-aged marathoners. This finding is interesting and may be related to differences in lifestyle factors (e.g., caffeine, alcohol, and electronic device use), as well as relationships with modern technology (i.e., sleep tracker use). In *post hoc* analyses, we explored whether these age groups differed across daily caffeine use, weekly alcohol use, frequency of electronic device use within 1 h before bedtime, and whether or not they used a sleep tracker, since these variables can potentially influence SOL. No significant differences between groups were observed for weekly alcohol use or likelihood of being a sleep tracker user. However, the results suggested that young adult marathoners utilized an electronic device within 1 h before bedtime significantly more frequently than middle-aged marathoners (β = 0.27; *p* < 0.0001; 95% CI = [0.14, 0.39]) and that middle-aged adults consumed more caffeine than younger adult marathoners (β = 0.27; *p* < 0.0001; 95% CI = [0.33, 0.61]). The caffeine finding is surprising, given that increased caffeine consumption can be associated with worse sleep initiation abilities [23]. Yet, the timing of caffeine consumption plays a key role in this relationship [24], and we were unable to account for that based on the collected information. With that said, the differences across age groups in electronic device use within 1 h before bedtime is interesting in the context of the increased SOL associated with young adults, relative to middle-aged adults. It is very plausible that this lifestyle characteristic contributes to longer SOL (further discussed in the next section) and, thus, degraded sleep health. As such, increased attention to electronic device use within 1 h of bedtime within younger marathoners may be warranted.

### 4.3. Weekly Frequency of Electronic Device Use and Prolonged Sleep Onset Latency (SOL) Were Associated with Worse Marathon Completion Time

When evaluating the relations of sleep health characteristics and lifestyle factors with marathon completion time, only SOL and weekly frequency of electronic device use within 1 h of bedtime (pre-bedtime electronic device use) were significantly associated with marathon completion time in both unadjusted and adjusted analyses. These relationships suggested that longer SOL and more frequent pre-bedtime electronic device use are associated with longer (“worse”) marathon completion time. Importantly, in analyses assessing relations between lifestyle factors and sleep health characteristics, weekly pre-bedtime electronic device use was significantly associated with longer SOL. In *post hoc* analyses, we explored a potential mediating role of either variable on the other’s relationship with marathon completion time. In a regression model that regressed marathon completion time on both SOL and weekly pre-bedtime electronic device use, significance was still observed for both SOL (*p* = 0.004) and weekly pre-bedtime electronic device use (*p* = 0.01). Although both relationships were weakened, relative to the unadjusted analyses related to marathon completion time, the relationship for weekly pre-bedtime electronic device use was notably more weakened, relative to SOL (which was only slightly weakened). Taken together, our results suggest that both SOL and frequency of pre-bedtime electronic device use are important factors to consider for marathoners in the context of marathon completion time. Yet, these results also may suggest that different types of electronic device use have differential influences on SOL, given that the relationship between weekly frequency of electronic device use within 1 h of bedtime is notably weakened when controlling for SOL. Thus, a fruitful line of research may be to explore which types of electronic devices (e.g., social media, TV/streaming, listening to music, reading on a tablet, etc.) most principally contribute to SOL difficulties in marathoners. Advancing this line of inquiry would be beneficial for best practice guidelines that are not only designed to enhance the performance of marathoners, but also their overall well-being.

### 4.4. Sleep Tracker Use Associates with Worse Sleep Satisfaction: Orthosomnia, Inaccurate Feedback, or Both?

We also observed a significant relationship between sleep tracker use and perception of sleep quality/satisfaction across the marathoner sample. The results suggested that those who used a sleeper tracker experienced lower levels of subjective sleep satisfaction. When considering this finding, two relevant, overlapping potential explanations come to mind. First, these individuals may have had “orthosomnia” tendencies, whereby they were putting greater efforts into their sleep that often results in degraded sleep quality. Considering that marathoners (and endurance athletes broadly) often are hyperattentive to training and lifestyle details, these individuals may be at heightened risk for orthosomnia tendencies [19]. Yet, it is also possible that the utilized devices were providing inaccurate estimations of sleep quantity and quality, which could have negatively biased these marathoners’ perceptions of sleep quality. Given that this study relied on data collected in 2016, when commercially available sleep tracking technology had major shortcomings in sleep quantification and classification abilities [25], it is not just possible but likely that these athletes were being provided inaccurate sleep feedback [26]. Ultimately, these potential explanations could be intersecting, whereby sleep tracker users may have had a default propensity for heightened attention to their sleep (aligning with the orthosomnia hypothesis), while also being provided inaccurate sleep feedback that has the potential to negatively bias perception of sleep quality. As such, marathoners (and athletes in general), coaches, and attending providers need to be mindful of the athlete’s psychological relationship with sleep health and tools relevant to sleep health, as over-fixation can contribute to deleterious outcomes for sleep and, thus, performance and overall well-being.

## 5. Limitations

Despite the many strengths of this investigation, there are also limitations that warrant attention. First, although the use of the ASSQ is a strength of this investigation, we still needed to rely upon subjective, retrospective data. Future research designs assessing sleep in marathoners should aim to use objective measures, when viable. Additionally, this study exclusively focused on marathoners from the 2016 London Marathon, which limits the generalizability of the findings across other marathons and marathoners. These data were also reflective of marathoners preceding competition. As such, the collected sleep and lifestyle variables may not translate to other periods of life for the marathoners. We also were limited in our ability to capture diverse, individual characteristics, which further limits the generalizability of the findings. This also limited the scope of relations between sleep health and individual characteristics that we could evaluate, as well as inhibited further ability to account for certain individual characteristics (e.g., psychological factors) that are likely to play key roles in the observed relationships with sleep health and marathon completion time. Furthermore, the self-reported, binary male/female characterization does not afford the ability to appropriately capture non-binary or transgender marathoners. We also acknowledge that our assumption of this response representing biological sex may be a limitation of the methodology. Additionally, we were unable to characterize the marathoners based on whether they qualified for the event or received an entry through the lottery system, which limited our ability to classify runner ability (i.e., elite vs. recreational). We also recognize the potential of certain selection biases for our sample, including the fact that there may be important differences between those who chose to participate in the data collection during registration versus those that declined. Additionally, despite a relatively large sample of marathoners, the sample only reflects a relatively small percentage of the overall London Marathon cohort. We also did not have data from marathoners who did not complete the race, yet this was a very small percentage of the overall London Marathon cohort. There are also a multitude of other, relevant variables that may impact sleep and marathon performance, such as overall physical and psychological health, training regimes, dietary habits, and social, familial, occupational, and academic factors, that we were unable to account for due to the limited scope of data collection.

## 6. Conclusions

This investigation was the first to evaluate the relationships between sleep health, individual characteristics, lifestyle factors, and marathon completion time in marathoners. The ability to capture sleep health using the Athlete Sleep Screening Questionnaire (ASSQ) is a major strength of the investigation, along with the relatively large sample of marathoners (N = 943). Within the sample, 23.5% of marathoners reported overall sleep difficulties on the ASSQ at a severity warranting follow-up with a trained sleep provider. Generally, middle-aged adult marathoners reported worse sleep health characteristics, relative to young adult marathoners. However, young adult marathoners did report significantly longer sleep onset latency (SOL), which may be at least partially explained by their significantly more frequent weekly use of electronic devices within 1 h before bedtime. Increasing weekly frequency of electronic device use and longer SOL emerged as predictors of longer (“worse”) marathon completion time, and our results suggest the need for future research to more carefully evaluate relations between the types of electronic device used in the hour preceding bedtime, SOL, and marathon completion time. Lastly, sleep tracker users reported significantly worse sleep satisfaction, relative to non-users, which could reflect orthosomnia tendencies in this population, perception bias influenced by inaccurate device feedback, or both. Going forward, it will be key for marathoners (and athletes in general), coaches, and providers to be mindful of the athlete’s psychological relationship with sleep health and tools relevant to sleep health, as over-fixation has the potential to contribute to deleterious outcomes for sleep health and, thus, performance and overall well-being.

## Figures and Tables

**Table 1 brainsci-13-01346-t001:** Descriptive Statistics across Analytic Sample.

SAMPLE CHARACTERISTICS																	
**Sample Size (N)**									943								
**Percent Male (%)**									64.5%								
**Percent Female (%)**									35.5%								
**Marathon Completion Time (minutes)**									251 ± 53.3								
**Age Group Percentage: Based on London Marathon Designation**																	
18–39 years		40–44 years				45–49 years			50–54 years			55–59 years				60–64 years	
66.5%		15.5%				10.1%			5.41%			1.48%				0.95%	
**Age Group Percentage: Statistical Analyses**																	
Young Adults (18–39 years)									Middle-aged Adults (40–64 years)								
66.5%									33.5%								
**Runner Performance Relative to “Good for Age (GFA)” Qualifying Threshold**																	
Faster Than GFA (GFA-Faster)									Slower Than GFA (GFA-Slower)								
21.4%									78.6%								
**SLEEP HEALTH CHARACTERISTICS**																	
**Global Sleep Difficulty Score (SDS): Severity Characterization**																	
None				Mild					Moderate					Severe			
35.6%				40.8%					18.1%					5.41%			
**Total Sleep Time (TST)**																	
<5 h		5–6 h				6–7 h			7–8 h			8–9 h				>9 h	
2.97%		17.9%				40.2%			33.0%			5.83%				0.11%	
**Sleep Satisfaction**																	
Very Dissatisfied			Somewhat Dissatisfied				Neutral				Somewhat Satisfied				Very Satisfied		
1.91%			19.7%				15.5%				47.7%				15.2%		
**Sleep Onset Latency**																	
<15 min				16–30 min					31–60 min					>60 min			
53.9%				32.2%					11.7%					2.23%			
**Sleep Maintenance Issues**																	
None				1–2x per week					3–4x per week					5–7x per week			
36.8%				44.6%					11.6%					7.00%			
**Sleep Medication Use**																	
None				1–2x per week					3–4x per week					5–7x per week			
94.3%				4.88%					0.21%					0.64%			
**LIFESTYLE FACTORS**																	
**Electronic Device Use within One Hour of Bedtime**																	
1–3x per week			4–6x per week				Everyday				Not At All				No Response		
15.8%			19.9%				58.0%				5.20%				1.06%		
**Sleep Tracker User**																	
Yes									No								
11.6%									88.4%								
**Caffeine Use (Per Day)**																	
<1 per day		1–2 per day				3 per day			4 per day			5+ per day				No Response	
18.9%		32.9%				22.3%			13.3%			0.00%				12.7%	
**Alcohol Use (Per Week)**																	
Does Not Drink	<2 drinks				2–4 drinks			5–7 drinks		8–14 drinks			>14 drinks				No Response
14.4%	24.4%				27.3%			20.4%		9.86%			3.08%				0.64%

**Table 2 brainsci-13-01346-t002:** Relations of Sex, Gender, and Runner Performance with Sleep Characteristics.

Sex							
Sleep Characteristic	Male	Female	β	SE	*p* Value	_p_η^2^	95% CI [LL, UL]
SDS	5.80 (2.49)	5.79 (2.82)	−0.01	0.02	0.92	0.000	[−0.37, 0.33]
TST	2.83 (0.85)	2.63 (0.84)	−0.20	0.06	0.001	0.011	[−0.30, −0.07]
Satisfaction	1.45 (1.02)	1.47 (1.06)	0.02	0.07	0.71	0.000	[−0.11, 0.16]
SOL	0.63 (0.76)	0.62 (0.81)	−0.01	0.05	0.83	0.000	[−0.12, 0.09]
Maintenance	0.85 (0.85)	0.96 (0.89)	0.11	0.06	0.08	0.003	[−0.01, 0.22]
Medication Use	0.05 (0.29)	0.11 (0.39)	0.06	0.02	0.03	0.005	[0.01, 0.09]
**Age**							
Sleep Characteristic	Young Adults	Middle-Aged Adults	β	SE	*p* value	_p_η^2^	95% CI [LL, UL]
SDS	5.67 (2.62)	6.05 (2.59)	0.38	0.20	0.04	0.005	[0.02, 0.73]
TST	2.70 (0.87)	2.89 (0.81)	0.19	0.06	0.001	0.011	[0.08, 0.31]
Satisfaction	1.38 (0.99)	1.60 (1.09)	0.22	0.07	0.003	0.009	[0.07, 0.35]
SOL	0.69 (0.80)	0.48 (0.71)	−0.21	0.05	< 0.0001	0.017	[−0.32, −0.11]
Maintenance	0.83 (0.85)	1.00 (0.90)	0.17	0.06	0.004	0.009	[0.06, 0.29]
Medication Use	0.07 (0.31)	0.08 (0.37)	0.01	0.02	0.50	0.000	[−0.03, 0.06]
**Runner Performance (Relative to GFA)**							
Sleep Characteristic	GFA-Faster	GFA-Slower	β	SE	*p* value	_p_η^2^	95% CI [LL, UL]
SDS	5.76 (2.59)	5.92 (2.70)	0.16	0.20	0.45	0.001	[−0.25, 0.57]
TST	2.75 (0.85)	2.80 (0.85)	0.05	0.07	0.42	0.001	[−0.08, 0.19]
Satisfaction	1.44 (1.03)	1.52 (1.03)	0.08	0.08	0.35	0.001	[−0.08, 0.24]
SOL	0.65 (0.78)	0.53 (0.77)	−0.12	0.06	0.06	0.004	[−0.24, 0.00]
Maintenance	0.86 (0.83)	1.00 (0.99)	0.14	0.07	0.04	0.005	[0.01, 0.28]
Medication Use	0.07 (0.32)	0.07 (0.36)	0.00	0.03	0.92	0.000	[−0.05, 0.05]

**Table 3 brainsci-13-01346-t003:** Relations of Lifestyle Factors with Sleep Characteristics.

Sleep Tracker User							
Sleep Characteristic	No	Yes	β	SE	*p* Value	_p_η^2^	95% CI [LL, UL]
SDS	5.75 (2.60)	6.08 (2.62)	0.34	0.26	0.20	0.002	[−0.18, 0.86]
TST	2.74 (0.85)	2.88 (0.86)	0.14	0.09	0.11	0.003	[−0.03, 0.31]
Satisfaction	1.43 (1.01)	1.64 (1.11)	0.22	0.10	0.04	0.005	[0.01, 0.42]
SOL	0.63 (0.79)	0.55 (0.70)	−0.08	0.08	0.31	0.001	[−0.24, 0.07]
Maintenance	0.88 (0.85)	0.92 (0.93)	0.04	0.09	0.66	0.000	[−0.13, 0.21]
Medication Use	0.07 (0.33)	0.09 (0.37)	0.02	0.03	0.51	0.000	[−0.04, 0.09]
**Daily Caffeine Use**							
Sleep Characteristic	β			SE	*p* value	_p_η^2^	95% CI [LL, UL]
SDS	0.17			0.09	0.06	0.004	[−0.01, 0.36]
TST	0.10			0.03	0.002	0.012	[0.04, 0.15]
Satisfaction	0.05			0.04	0.18	0.002	[−0.02, 0.12]
SOL	−0.01			0.03	0.66	0.000	[−0.07, 0.04]
Maintenance	0.03			0.03	0.29	0.001	[−0.03, 0.09]
Medication Use	0.01			0.01	0.53	0.000	[−0.02, 0.03]
**Weekly Alcohol Use**							
Sleep Characteristic	β			SE	*p* value	_p_η^2^	95% CI [LL, UL]
SDS	−0.01			0.07	0.86	0.000	[−0.14, 0.12]
TST	0.03			0.02	0.23	0.002	[−0.02, 0.07]
Satisfaction	−0.01			0.03	0.73	0.000	[−0.06, 0.04]
SOL	−0.03			0.02	0.10	0.003	[−0.07, 0.01]
Maintenance	−0.01			0.02	0.75	0.000	[−0.05, 0.04]
Medication Use	0.01			0.01	0.18	0.002	[−0.01, 0.03]
**Weekly Electronic Use Within 1 Hour of Bedtime**							
Sleep Characteristic	β			SE	*p* value	_p_η^2^	95% CI [LL, UL]
SDS	0.05			0.09	0.56	0.000	[−0.13, 0.23]
TST	0.00			0.03	0.94	0.000	[−0.06, 0.06]
Satisfaction	0.01			0.04	0.75	0.000	[−0.06, 0.08]
SOL	0.06			0.03	0.03	0.005	[0.00, 0.11]
Maintenance	−0.03			0.03	0.31	0.001	[−0.09, 0.03]
Medication Use	0.01			0.01	0.28	0.001	[−0.01, 0.04]

**Table 4 brainsci-13-01346-t004:** Relations of Sleep Characteristics and Lifestyle Factors with Marathon Completion Time.

Sleep/Lifestyle Variable	β	SE	*p* Value	_p_η^2^	95% CI [LL, UL]	*p* Value (Adjusted)	_p_η^2^ (Adjusted)
SDS	1.04	0.66	0.12	0.003	[−0.26, 2.35]	0.05	0.004
TST	1.23	2.04	0.55	0.000	[−2.78, 5.24]	0.04	0.004
Satisfaction	0.90	1.69	0.60	0.000	[−2.41, 4.21]	0.43	0.001
SOL	6.86	2.23	0.002	0.010	[2.49, 11.2]	0.004	0.009
Maintenance	0.91	2.00	0.65	0.000	[−3.02, 4.85]	0.81	0.000
Medication Use	4.03	5.25	0.44	0.001	[−6.27, 14.3]	0.85	0.000
Weekly Electronic Device Use	5.20	1.88	0.006	0.008	[1.51, 8.88]	0.03	0.005
Sleep Tracker Use	1.72	5.42	0.75	0.000	[−8.92, 12.4]	0.97	0.000
Daily Caffeine Use	−4.48	1.88	0.02	0.007	[−8.17, −0.80]	0.19	0.002
Weekly Alcohol Use	−3.49	1.33	0.009	0.007	[−6.10, −0.89]	0.15	0.002
Weekly Alcohol Use (HAC vs. Other)	−18.54	10	0.07	0.004	[−38.2, 1.16]	0.17	0.002

## Data Availability

The data presented in this study are available based on reasonable request to the corresponding author.

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
