# Peer review of "Sleep Health, Individual Characteristics, Lifestyle Factors, and Marathon Completion Time in Marathon Runners: A Retrospective Investigation of the 2016 London Marathon"

_brainsci, 2023, doi:10.3390/brainsci13091346_

Round 1
Reviewer 1 Report
Dear Authors,
I have read your paper with interestas relation between physical activity, training and sleep are very important. The paper have some disadvantages that hopefully may be fixed.
1. The aim of the study is not precise. You have written: "Therefore, this investigation represents the first of its kind with the 95
aim of exploring the sleep characteristics of a large sample of marathon runners with a validated screening tool designed for the unique athletic population. " . In fact You have collected data upon sleep in marathon runners in period just before an important event. Preparations to that event, anticipation stress, intense training might have impacted the results, therefore upon this study it is impossible to diagnose runners as good or bad sleepers.
2. Did you collect any more information about participants of study (their health, any chronic conditions, intensity of trainings etc.) If so - present them
3. You should clearly present the SDS scores in the different age groups
4. I think that excluding the ones who did not finish the marathon was not a good idea. Maybe the did not finish because of worse sleep? That would be interesting. Do you have their SDS scores? If so, please compare them with the scores of those who finished the marathon.
5. Do you have data allowing comparison of quality of sleep and results in Marathon? That would be a valuable analysis. (answer to question if good sleep prior an event make a runner more efficient)
6. Disussion should be less speculative. E.g., explainig the difference in SDS between genders by difference in prevalence of RLS is very superficial.
Summing up - presented data does not provide any sound information about sleep of marathon runners. Additional data should be presented to make the picture more complete.
English language requires very minor improvement.
Author Response
see attchment

Reviewer 2 Report
Thank you for allowing me to review your manuscript. Despite the importance of the topic, the manuscript and information needs to be better presented. I have several concerns that I would like you to address before the manuscript is ready for publication. Some of my comments pertain to the structure and organization of the paper, while others relate to the need for additional information and clarity on specific topics.
I am not sure if these are the most appropriate keywords, please, consider thinking about that. The keywords will help the readers to find your research.
The introduction is well-structured; however, some adjustments are necessary. For instance, while the authors discuss the importance of sleep for sports performance, it would be crucial to include evidence regarding the specific relationship between sleep and marathoners. Additionally, emphasizing the significance of this topic for non-professional runners is important, and the presentation of concrete evidence concerning the positive and negative effects on both health and performance in this population would greatly enhance the introduction.
The information about the questionnaire used in this study could be moved to the methods section. Alternatively, the authors should delve deeper into the measurement of sleep and be more specific about the study's purpose. Merely stating the purpose as "to explore" sleep characteristics lacks informativeness and may also create a mismatch with other parts of the manuscript."
L100 – “The ASSQ that included assessment of lifestyle habits that could cause poor sleep 100 was administered to participants of the 2016 London Marathon”. Also, ensure that the work follows the guidelines for the study design (i.e., STROBE, for example, if the authors consider the study as a cross-sectional study).
The methods section should begin by outlining the study design and approach used, as well as the eligibility criteria for study participation. Additionally, it is essential to explain why the study was specifically limited to the London Marathon. Ethical aspects of the research, such as ethical approval and informed consent from participants, should be clearly addressed.
For information regarding age groups, please refer to the Virgin London Marathon organizing committee as a source.
To ensure clarity, the authors should provide a more explicit definition of participants' running ability. Terms like "fast" or "regular" are subjective and lack consensus in runner classification. External validation of the study would benefit from a clearer explanation of these categories.
The sample included in this study represents what percentage of the total marathoners?"
"For the statistical analysis, I suggest starting with reporting the descriptive information, followed by the comparative analysis. It would be important for the authors to test the normality of the data and examine the assumptions for the MANOVA. Please provide this information along with the procedures followed to address any non-compliance with assumptions.
Regarding the use of ANOVA, the authors mentioned employing it to test for significant variance in sleep outcomes across predictor variables (gender, runner ability, and age group). However, it is unclear how the model was run since the authors referred to "predictors" and outcome variables. To enhance the analysis, I recommend including a regression model after the ANOVA and MANOVA analysis, considering the reported results.
Since sex and age are considered as factors in the statistical analysis, please adjust the descriptive information and include stratification by these groups in Table 1, along with the information for the total sample.
For the total sleep time, it is essential to clarify how the categories are mutually excluded. For instance, are those reporting 6 hours classified among 5 to 6 hours or 6 to 7 hours?
The tables reporting MANOVA results need to be better organized for ease of understanding. Consider providing a more descriptive title or utilize figures to present the main results, which may enhance clarity.
Please maintain consistency in the use of decimal places throughout the manuscript, and also regarding the term "predictors" to avoid confusion."
L270 – “Overall, these results show that sleep disturbances are present within a highly trained and physically fit population focused on optimal athletic performance, whereby sleep health should play a central role”. To ensure that, the authors need to be more specific about the sample characteristics (performance). In addition, most of the participants were classified as “regular”, not necessarily, being related to the authors' words.
Regarding the study limitations, it is important to include the lack of information about the participants' training background, as well as other potential factors that could influence the results. Additionally, please discuss any other potential biases present in the study and how these biases may impact the external validation of the findings.
Minor editing of the English language required
Round 2
Reviewer 1 Report
Dear Authors,
I find the manuscript noticeably improved therefore I find it suitable for publication.
Reviewer 2 Report
Well done with the changes.
Minor editing of English language required